# Modeling reveals a metabolic basis of competition among *Dehalobacter* strains during tandem chloroform and dichloromethane metabolism

Olivia Bulka,[1] Elizabeth A. Edwards,[1] Radhakrishnan Mahadevan[1]

**ABSTRACT**  SC05-UT is an anaerobic, heterogenous microbial enrichment culture that reduces chloroform to dichloromethane through reductive dechlorination, which it further mineralizes to carbon dioxide. This dichloromethane mineralization yields electron equivalents that are used to reduce chloroform without the addition of exogenous electron donor. By studying this self-feeding chloroform-amended culture and a dichloromethane-amended enrichment subculture (named DCME), we previously found the genomic potential to perform both biodegradation steps in two distinct *Dehalobacter* strains: *Dehalobacter restrictus* SAD and *Candidatus* Dehalobacter alkaniphilus DAD. Though present in each enrichment culture, strain SAD is more abundant in the chloroform-fed subculture SC05-UT, while strain DAD is more prominent in the dichloromethane-fed subculture DCME. To understand if genomic differences between strains impact their metabolic mechanisms, the genome of each strain was curated to reconstruct genome-scale metabolic models of each strain, which were then constrained based on thermodynamic and experimental conditions. We demonstrate that metabolic differences between the two strains may allow *Dehalobacter* strain DAD to outcompete strain SAD in the absence of chloroform, while strain SAD exhibits an advantage in the presence of chloroform. Additionally, we predict electron cycling methods to reconcile cellular redox imbalances during tandem chloroform and dichloromethane dechlorination. This work highlights the importance of hydrogen and amino acid exchange in these microbial communities and contributes to the growing body of work surrounding organohalide syntrophy.

**IMPORTANCE**  Chloroform and dichloromethane contaminate groundwater around the world but can be remediated by microbes capable of metabolizing these toxic compounds. Here, we study two distinct strains of *Dehalobacter* and show that while both strains can degrade both chloroform and dichloromethane, differences in their genetic makeup allow each strain to thrive under different environmental conditions. This has implications for understanding the fate of halogenated methanes in the environment and the application of *Dehalobacter* for bioremediation of chlorinated compounds.

**KEYWORDS**  dehalogenation, *Dehalobacter*, metabolic modelling, organohalide

The bioremediation industry has long harnessed the power of natural communities for dechlorination of organohalides. Many of these communities contain *Dehalobacter* spp., a key genus for reductive dechlorination of chloroalkanes and alkenes. Despite experimental characterization of more than 40 strains of *Dehalobacter*—and identification of many more strains by 16S amplicon sequencing alone—only six closed genomes have been published within this genus (1–5). Recent pangenomic and phylogenetic work has divided the *Dehalobacter* genus into multiple species:

Address correspondence to Radhakrishnan Mahadevan, krishna.mahadevan@utoronto.ca.

The authors declare no conflict of interest.

See the funding table on p. 19.

  10.1128/msystems.00847-25  1

*Dehalobacter restrictus*, *Candidatus* Dehalobacter alkaniphilus, and *Candidatus* Dehalobacter aromaticus (5). *Dehalobacter* are known to be fastidious anaerobic organisms, with fragmented TCA cycles and a reliance on exogenous amino acids and other organic acids (6, 7). Only *Dehalobacter restrictus* has been successfully isolated and deposited in a culture collection, further complicating experimental and computational study of their metabolisms (7–10).

Two genome-scale metabolic models representing organohalide-respiring bacterial genera have been reconstructed: a *Dehalococcoides* pangenome model (*i*AI549 [11]) and a *Dehalobacter* pangenome model (*i*HH623 [12]). The *Dehalobacter* model was initially created from the genome of *Ca*. D. alkaniphilus CF and curated to encompass the metabolism of the genus as a whole using annotations from the available genomes of strains CF, DCA, PER-K23, E1, and UNSWDHB (12, 13). Though it has been widely accepted that *Dehalobacter* strains have a very restricted metabolism—each reducing a chlorinated electron acceptor with $H_2$ as an electron donor—recent discoveries within the genomes of two novel *Dehalobacter* strains from a mixed microbial culture (SC05) suggest more metabolic variability across species than was previously thought (14).

SC05 is a mixed microbial anaerobic chloroform (CF)-degrading enrichment culture originally sampled in 2010 from a site polluted with chlorinated ethenes and ethanes (Fig. 1A) (15, 16). Through reductive dechlorination, it reduces CF to dichloromethane (DCM), which it further mineralizes to carbon dioxide and hydrogen (16). This hydrogen can then act as an electron donor to support CF reduction through "self-feeding," allowing a subculture (SC05-UT) to continually dechlorinate CF to DCM without feeding any exogenous electron donor since 2018 (15). An additional subculture, named DCME, was established with DCM amendment alone since 2019 (15). Though both of these heterogeneous subcultures consist of many microbial species, only a single *Dehalobacter* ASV was detected through 16S amplicon sequencing since their conception; it flourished both during CF dechlorination and DCM mineralization, which positioned it as the sole dechlorinator in the SC05 culture (15).

When the first metagenomes of the two SC05 subcultures (SC05-UT and DCME) were assembled, two distinct *Dehalobacter* metagenome-assembled genomes (MAGs) emerged, one from each metagenome (14). Each MAG was searched genomically and proteomically for the enzymes known to perform CF dechlorination—reductive dehalogenases (RDases)—and DCM mineralization—methyltransferases encoded by the DCM catabolism (*mec*) cassette. Only one RDase was expressed in each culture; it was named AcdA in SC05-UT and was shown to dechlorinate CF to DCM (14). The *mec* cassette was also highly expressed in both cultures (14), becoming one of only two reports to ascribe DCM metabolism to *Dehalobacter* (15, 17).

Through additional sequencing, these MAGs were curated into two distinct closed genomes belonging to different *Dehalobacter* species: *D. restrictus* SAD and *Ca*. D. alkaniphilus DAD (5, 18). Both genomes share the RDase–*mec* cassette gene neighborhood, a region with 98.7% sequence identity across *Dehalobacter* species, in an integration "hot spot" detected in the *Dehalobacter* pangenome (5). Though encoding nearly identical functional genes, these strains each dominate the culture under different conditions—strain SAD is abundant in CF-fed SC05-UT, while strain DAD is more prominent in the DCM-fed DCME (5). Despite this differential predominance, their mutual presence in both cultures challenges the assumption that only one strain is actively growing in each subculture (5).

Here, we aim to quantify these *Dehalobacter* strains in each subculture and rationalize their varying predominance using experimental data coupled with metabolic models, which can be optimized to predict the growth potential of each *Dehalobacter* strain under varying experimental conditions in SC05-UT and DCME. Furthermore, we seek to identify differences in metabolic potential across these two *Dehalobacter* species to determine ecological significance.

## RESULTS AND DISCUSSION

### Each *Dehalobacter* strain performs two remediation steps

Strain-specific growth was measured during each remediation step in replicate SC05-UT and DCME subcultures. All bottles were initially fed CF and hydrogen, and *Dehalobacter* growth was quantified during the sequential degradation of CF and DCM using qPCR to target a divergent, single-copy core gene (*flgC* [15]). SC05 can grow by three growth modes (Fig. 1A)—Mode 1: supplied $H_2$ as an electron donor, CF as an electron acceptor; Mode 2: supplied DCM as electron donor, which provides electrons to reduce CF as the ultimate electron acceptor; and Mode 3: supplied DCM as electron donor for mineralization to $CO_2$ with $H^+$ as the electron acceptor. Under normal maintenance, DCME usually operates in Mode 3, as it is typically only fed DCM; SCO5-UT usually operates in Mode 2, except when inoculating a new bottle (Fig. 1A).

In the three SC05-UT bottles, only strain SAD grew during CF dechlorination (Fig. 1D). The measured cell yield (Table 1) is within the error margin of previously quantified SC05 *Dehalobacter* yields when growing on CF alone and 4-fold to 10-fold higher than that of other CF-dechlorinating strains of *Dehalobacter* (15, 19–22). Some DCM produced from CF dechlorination was consumed concurrently, which may account for the higher yield (Fig. 1B). Strain DAD did not grow during this period, which shows unequivocally that strain SAD is the main CF dechlorinator in SC05-UT. During DCM mineralization (Mode 3), strain DAD did not grow in two of three replicates, but in one replicate (R2), strain DAD increased almost 100-fold (Fig. 1D; Table 1). Although demonstrating that both strains are independently able to grow by DCM mineralization, this outcome diversity suggests that DCM mineralization may at times be a joint effort between the two *Dehalobacter* strains in SC05-UT.

In the DCME sub-transfers, *Dehalobacter* strain DAD more than doubled when CF was dechlorinated from days 7 to 12 (Fig. 1E; Table 1). No DCM was degraded at this time, confirming that strain DAD can dechlorinate CF using $H_2$ as an electron donor (Mode 1). DCM mineralization yielded fewer strain DAD cells in the DCME replicates than previously calculated ([1.23 ± 0.65] × $10^{13}$ cells/eeq donor [15]), but notably, its yield in SC05-UT R2 is cohesive with the previous work (Table 1). After day 12, though *Dehalobacter* strain SAD was initially 100-fold less abundant than strain DAD in the DCME sub-transfers, strain SAD also increased as CF was dechlorinated until day 19 (Fig. 1E, orange). Because strain DAD also grew during this period, their yields cannot be disentangled. Nonetheless, strain SAD's rapid growth, despite low relative abundance, suggests that it may dechlorinate CF faster or more efficiently than strain DAD, providing an explanation for outcompetition when SC05-UT is maintained on CF alone.

### Distinct *Dehalobacter* species metabolic models are constructed for each strain

Species-specific genome-scale metabolic models were reconstructed to describe *Dehalobacter* strains SAD and DAD (Table 2) and curated through comparative genomics with support from proteomics (described in Texts S1 and S2, respectively). Organohalide metabolism of each *Dehalobacter* strain was updated; reductive dechlorination of CF to DCM and DCM assimilation to methylene-tetrahydrofolate (THF) are each represented by one reaction (Table S4). A reaction was added to represent a ferredoxin-mediated complex I-like enzyme, reflecting recent work describing electron transfer in *Desulfitobacterium* (a closely related genus) and *Dehalobacter restrictus* (23). Both strains' genomes encoded a complete Wood-Ljungdahl pathway (WLP) and eight homologous hydrogenases (Table S5). These hydrogenases—labelled Hyd-1 through Hyd-8—comprise four metabolic reactions (Table S6; Fig. S3), as classified by HYDdb (24). One additional hydrogenase was detected in strain DAD (Hyd-9, Table S6), but its representative reaction is redundant. Of the hydrogenases, the same four were expressed in each of the strains' proteomes, corresponding to three metabolic reactions: two electron-bifurcating hydrogenases, one Ech-type hydrogenase, and a periplasmic uptake

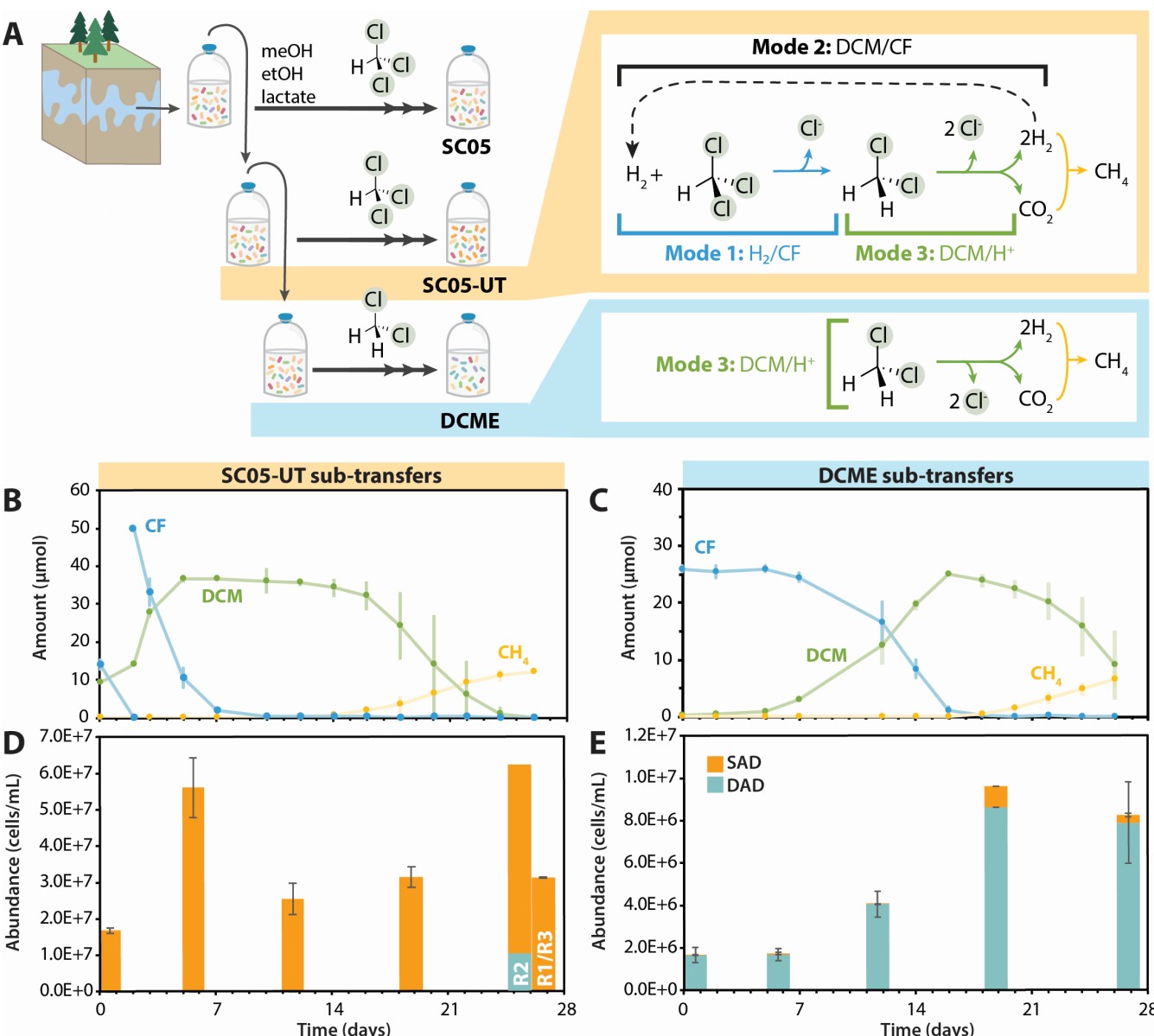

**FIG 1** Growth of *Dehalobacter* in SC05-UT and DCME, including (A) a schematic of culture provenance and metabolic modes and (B) dechlorination in SC05-UT sub-transfers and (C) DCME sub-transfers. (D and E) The *Dehalobacter* population in SC05-UT and DCME. SC05-UT: $n = 3$, error bars represent standard deviation; DCME: $n = 2$, error bars represent range. Day 26 samples from SC05-UT are separated by replicate (R1–R3) to showcase two outcomes.

hydrogenase (Table S6; Fig. S3). Genes encoding three unique metabolic reactions were detected in the genome of strain DAD and were added to the model accordingly: fumarate reductase, aspartate ammonia lyase, and a nitrogen fixation cassette (Fig. 2; Fig. S2). No unique metabolic genes were found in strain SAD.

## DCM is a less efficient electron donor than hydrogen

Because these strains grow under three different electron donor/acceptor modes, each model was curated using the thermodynamic constraints for each mode individually (Table 3; expanded upon in Text S1.3). The $H^+/e^-$ ratio was set to two, rendering the max $ATP/e^-$ ratio as 0.5 (Table S3). Energy transfer efficiency is calculated using $ATP/e^-$ ratios, which represent the amount of ATP produced per mole of electrons transferred from the electron donor (calculated by [model-predicted $ATP/e^-$]/[max $ATP/e^-$]). Energy

**TABLE 1** Calculated yields of each strain in SC05-UT and DCME subcultures[b]

| Subculture | Mode | Days | Amount degraded (mmol) | | Strain | Concentration (16S rRNA copies/mL) | | Yield (per eeq donor) | |
|---|---|---|---|---|---|---|---|---|---|
| | | | CF | DCM | | Initial | Final | 16S rRNA copies | Cells[a] |
| SC05-UT | 1 | 0–5 | 0.053 | 0.016 | SAD | $1.68 \times 10^7$ | $5.61 \times 10^7$ | $(3.7 \pm 0.78) \times 10^{13}$ | $(9.3 \pm 1.9) \times 10^{12}$ |
| | | | | | DAD | $1.93 \times 10^5$ | $1.72 \times 10^5$ | No growth | No growth |
| | 3 | 12–26 | 0 | 0.036 | SAD | $2.54 \times 10^7$ | $3.90 \times 10^7$ | $(9.4 \pm 3.0) \times 10^{12}$ | $(2.4 \pm 0.75) \times 10^{12}$ |
| | | | | | DAD | $1.54 \times 10^5$ | $1.65 \times 10^5$ | No growth | No growth |
| SC05-UT (R2) | 3 | 12–26 | 0 | 0.037 | DAD | $1.58 \times 10^5$ | $1.02 \times 10^7$ | $6.8 \times 10^{12}$ | $2.30 \times 10^{12}$ |
| DCME | 1 | 5–12 | 0.009 | 0 | SAD | $4.86 \times 10^4$ | $4.03 \times 10^3$ | No growth | No growth |
| | | | | | DAD | $1.67 \times 10^6$ | $4.04 \times 10^6$ | $(1.3 \pm 0.37) \times 10^{13}$ | $(4.4 \pm 1.2) \times 10^{12}$ |
| | 3 | 18–26 | 0 | 0.015 | SAD | $9.92 \times 10^5$ | $3.61 \times 10^5$ | No growth | No growth |
| | | | | | DAD | $8.60 \times 10^6$ | $9.79 \times 10^6$ | $2.92 \times 10^{10}$ | $9.7 \times 10^9$ |

[a]Calculated using four copies of 16S rRNA per cell for strain SAD and three for strain DAD.
[b]Yield is displayed as mean ± error of biological replicates (DCME, $n = 2$ [range]; SC05-UT, $n = 3$ [standard deviation]; no error is shown where one replicate is available).

transfer efficiency is lower during Modes 2 and 3, where DCM assimilation occurs, than during Mode 1 (Table 3). Thermodynamically, DCM mineralization has a favorable ΔG, but the cell's mechanism of metabolism may be unable to harness this energy to its full extent. Computationally, the model is restricted to using the same reactions in Mode 2 as in Modes 1 and 3, and thus cannot harness all energy theoretically available. Physiologically, this may be due to these same enzyme mechanism limitations, but we lack clear experimental data from this case. Thus, it remains unknown whether this result is representative of a biological phenomenon or whether more complex constraints are required to capture the switch to Mode 2. Overall, efficiencies are higher during CF dechlorination than DCM mineralization, which is also reflected by experimentally determined yields (Table 1).

## Unexplained activity evidenced by Mode 3 redox imbalance

To further assess metabolic differences between these strains, flux balance analysis (FBA) was used to optimize each model, and solutions were explored under three metabolic modes. All FBA simulations are described and reported in Tables S10 and S11, with a summary of key fluxes in Table S12 (described in Text S3). During Mode 3, optimization of each model using DCM as a sole electron donor and carbon source was initially infeasible due to stoichiometric redox constraints preventing ATP production (Fig. 3), despite experimental growth of both strains under these conditions. To harness ATP synthase, one of two reactions must establish a proton motive force (Fig. 3): a complex I-like oxidoreductase (FDXMQpp) or an energy-conserving hydrogenase (HYDA_Ech). Computationally, neither translocation mechanism could occur in the inital models during Mode 3, as described below.

FDXMQpp requires an active menaquinol sink, which is limited to two possible reactions in the strain DAD model and only one in strain SAD. The RDase uses menaquinol to reduce CF to DCM in both strains, which is inherently inoperable without providing CF. In strain DAD, a putative fumarate reductase (FRD2) could use menaquinol to catalyze fumarate reduction to succinate, but strain SAD lacks this gene (Fig. S1).

**TABLE 2** Statistics for each curated genome-scale model

| Parameter | *Dehalobacter restrictus* SAD (*i*OB638) | *Ca*. Dehalobacter alkaniphilus DAD (*i*OB649) |
|---|---|---|
| Genes | 638 | 649 |
| Total reactions | 1,068 | 1,071 |
| Exchange reactions | 63 | 63 |
| Compartments | 3 | 3 |
| Metabolites | 952 | 952 |

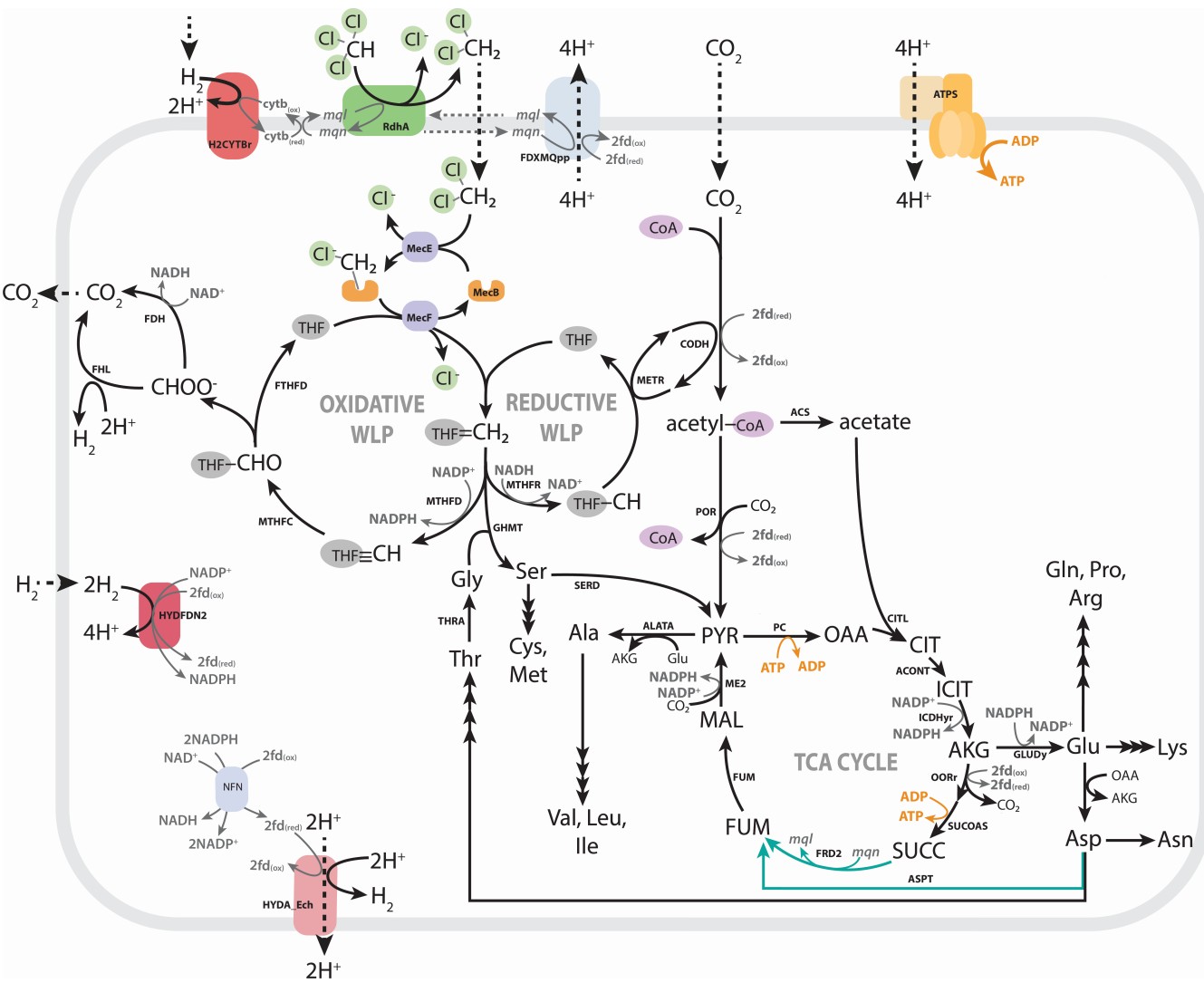

**FIG 2** Genetic landscape of central carbon and amino acid metabolism in *Dehalobacter* strains SAD and DAD. Teal pathways are encoded by strain DAD only (ASPT and FRD2). Metabolites and enzymes are abbreviated as follows: PYR, pyruvate; OAA, oxaloacetate; CIT, citrate; ICIT, isocitrate; AKG, alpha-ketoglutarate; SUCC, succinate; FUM, fumarate; MAL, malate; THF, tetrahydrofolate; MecEBF, *mec* cassette; MTHFD, methylene-THF dehydrogenase; MTHFC, methenyl-THF cyclohydrolase; FTHFD, formyl-TFH deformylase; FHL, formate hydrogen lyase; FDH, formate dehydrogenase; MTHFR, methylene-THF reductase; METR, 5-methyl-THF corrinoid/FeS protein methyltransferase; CODH, carbon monoxide dehydrogenase/acetyl-CoA synthase; ACS, acetate-CoA ligase; POR, pyruvate oxidoreductase; PC, pyruvate carboxylase; CITL, citrate lyase; ACONT, aconitate hydratase; ICDHyr, isocitrate dehydrogenase; OORr, oxoglutarate oxidoreductase; SUCOAS, succinate CoA ligase; FRD2, fumarate reductase; FUM, fumarate hydratase; ME2, malic enzyme; ALATA, alanine:oxoglutarate aminotransferase; GLUDy, glutamate dehydrogenase; ASPT, aspartate ammonia lyase; THRA, threonine acetaldehyde lyase; GHMT, methylene-THF:glycine hydroxymethyltransferase; SERD, serine ammonia lyase; NFN, electron-bifurcating transhydrogenase; HYDA_Ech/HYDFDN2/H2CYTBr/HYD-NADH, hydrogenases; FDXMQpp, complex I-like enzyme; RDase, reductive dehalogenase; ATPS, ATP synthase; WLP, Wood-Ljungdahl pathway; and TCA, tricarboxylic acid.

Consequently, FDXMQpp cannot reduce menaquinone and cannot function as a proton pump.

Alternatively, HYDA_Ech oxidizes ferredoxins to produce $H_2$ and translocate protons. Despite the presence of ferredoxin recycling reactions in the model, NADH accumulation precludes a feasible solution. Ferredoxins can be reduced via two main reactions: a Group A3 [NiFe] hydrogenase (HYDFDN2) that reversibly bifurcates (or confurcates) electrons between two equivalents of $H_2$ and one equivalent each of NAD(P)$^+$ and oxidized ferredoxins, and a ferredoxin-dependent transhydrogenase (NFN) that bifurcates electrons from two equivalents of NADPH to reduce oxidized ferredoxins

**TABLE 3** Theoretical thermodynamic constraints for energy metabolism of both *Dehalobacter* models[c]

| Strain | Growth mode (donor/acceptor) | ΔG (kJ/$e^-$)[b] | ATP/$e^-$ ratio | | | | H$^+$/$e^-$ ratio[a] | |
|---|---|---|---|---|---|---|---|---|
| | | | Therm. max | FBA max | | Energy transfer efficiency | Therm. max | FBA max |
| | | | | NGAM | Biomass | | | |
| SAD | 1 (H$_2$/CF) | −97.8 | 1.99 | 0.50 | 0.22 | 0.25 | 8.0 | 2.00 |
| DAD | | | | 0.50 | 0.29 | 0.25 | | 2.00 |
| SAD | 2 (DCM/CF) | −137.1 | 2.79 | 0.50 | 0.42 | 0.18 | 11.2 | 2.00 |
| DAD | | | | 0.50 | 0.34 | 0.18 | | 2.00 |
| SAD | 3 (DCM/H$^+$) | −80.6 | 1.64 | 0.06 | 0.12 | 0.04 | 6.6 | 0.25 |
| DAD | | | | 0.06 | 0.12 | 0.04 | | 0.25 |

[a]Assuming four protons are required per ATP synthesis by ATP synthase.
[b]Assumptions: [CF] = 0.001 M, [DCM] = 0.001 M, [Cl$^-$] = 0.033 M, [CO$_2$] = 0.2 atm, [H$^+$] = 1× 10$^{-7}$ M (pH 7), [H$_2$] = 1× 10$^{-4}$ atm, Temp = 298 K, $R$ = 0.008134.
[c]Therm. max = maximum calculated using thermodynamics; FBA max = maximum calculated using flux balance analysis; NGAM = [optimized for] non-growth associated maintenance; Biomass = [optimized for] biomass; Energy transfer efficiency = (model-predicted ATP/$e^-$)/(max ATP/$e^-$).

and NAD$^+$. Due to this strict stoichiometry and the absence of an NADH sink, especially in the absence of FDXMQpp, a feasible solution is unattainable during Mode 3.

Three strategies resolved the redox imbalance imposed by DCM assimilation. First, supplementation with additional carbon sources, such as amino acids, can absorb some of the accumulated reducing equivalents (strategy A, Fig. 3A). Alternatively, a hydrogenase or redox protein may produce H$_2$ using the accumulated NAD(P)H directly (strategy B, Fig. 3B), perhaps in conjugation with amino acid supplementation (strategy C).

## Strategy A: carbon source supplementation reconciles redox imbalance

The models were first supplemented with 5 mmol gdw$^{-1}$ day$^{-1}$ of each amino acid to reconcile the redox imbalance created during DCM assimilation and support biomass formation via conversion to pyruvate (Fig. 3A). This supplementation resulted in feasible growth solutions in both strains (Table S11), and the use of glutamate dehydrogenase (GLUDy) and alanine glyoxylate aminotransferase (ALATA) helped balance redox equivalents (Fig. S4). GLUDy and ALATA are in the top 10% of *Dehalobacter* proteins expressed in DCME and SC05-UT (Table S9). To produce biomass from 10 mmol gdw$^{-1}$ day$^{-1}$ DCM, strain SAD required at least 5 mmol gdw$^{-1}$ day$^{-1}$ of glutamate and serine as well as smaller amounts of glycine, glutamine, and aspartate (Table S12). Strain DAD could also incorporate 5 mmol gdw$^{-1}$ day$^{-1}$ aspartate due to its aspartate transferase gene (ASPT), which doubled the growth rate in this simulation (Table S12). This increased substrate flexibility allowed biomass production with lower uptake fluxes of amino acids overall compared to strain SAD (3 mmol) (Table S12; Fig. S4B).

In a complex community like SC05, metabolites are exchanged between diverse organisms. For example, several amino acid-recycling microbes (the Dehalobacteriia class and Spirochaetes) are prominent in DCME (15). The previously modeled *Ca*. D. alkaniphilus CF relied on many metabolites produced by microbes in its community to resolve a similar redox imbalance (6), and supplemental amino acids, yeast extract, or spent media have also aided isolation and enrichment of many *Dehalobacter* strains (6, 7, 9, 22). The heavier reliance of strain SAD on additional supplements for DCM mineralization may contribute to the prominence of strain DAD in DCME. While strain SAD is evidently competitive in CF-fed conditions, strain DAD can produce more biomass from DCM alone with less support from amino acid-producing microorganisms because of the increased ability to harness its incomplete TCA cycle.

## Strategy B: hydrogen as an alternative NAD(P)H sink

As an alternative to supplementation with external carbon sources, the wealth of hydrogenases in each *Dehalobacter* genome provides a putative sink for reducing equivalents. Both *Dehalobacter* genomes encode three homologous Group A3 [FeFe] hydrogenases: Hyd-3, Hyd-4, and Hyd-6 (Table S6), two of which were expressed via proteomics (Hyd-3 and Hyd-6). These hydrogenases' β-subunits exhibit two different

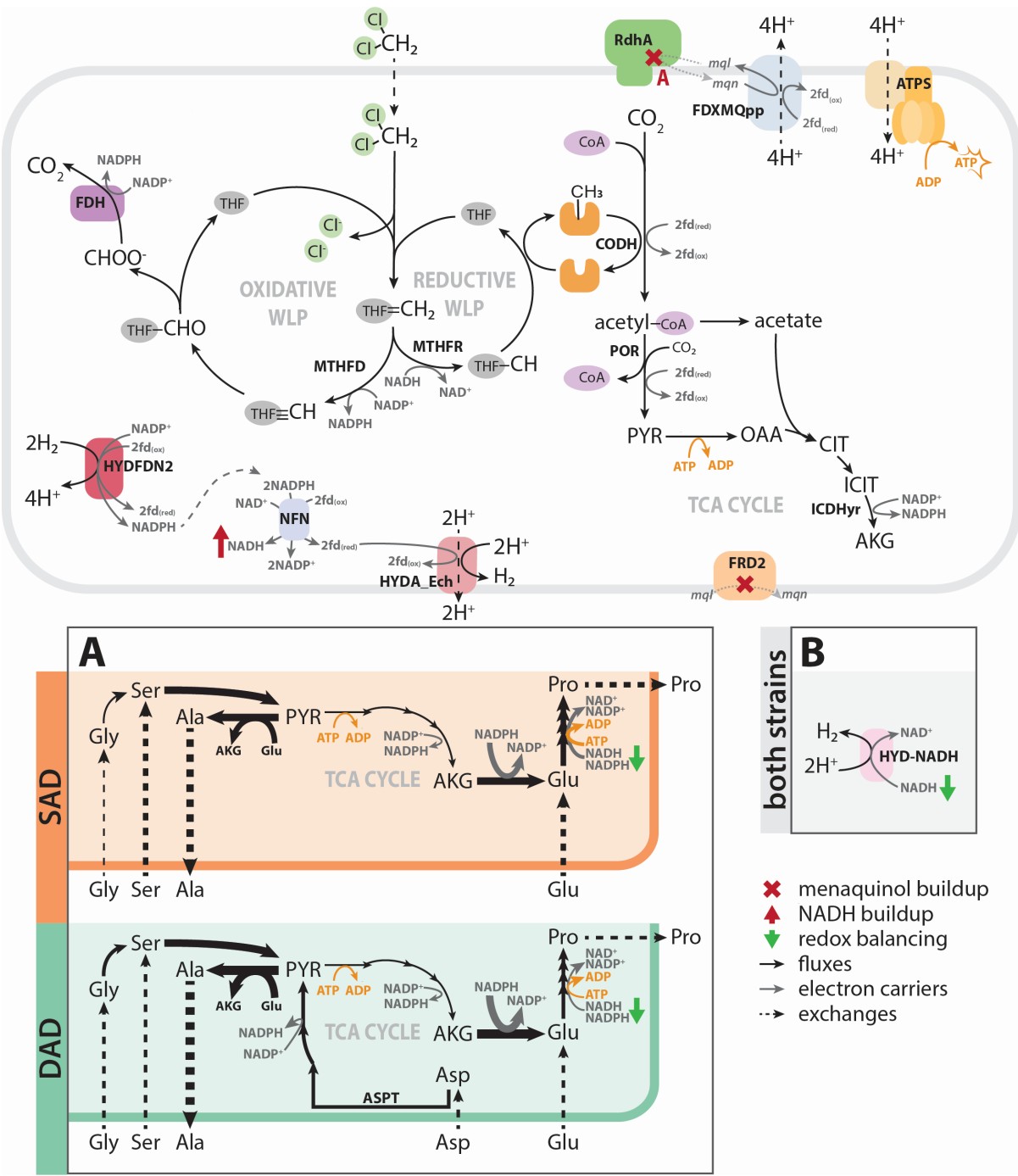

**FIG 3** Redox imbalance during Mode 3 and strategies to resolve it. Strategy (A) amino acid supplementation to balance redox cofactors in each strain. Strategy (B) NADH-dependent hydrogenase. PYR, pyruvate; ICIT, isocitrate; AKG, alpha-ketoglutarate; SUCC, succinate; FUM, fumarate; MAL, malate; THF, tetrahydrofolate; fdx, ferredoxin; mql, menaquinol; mqn, menaquinone; MTHFD, methylene-THF dehydrogenase; MTHFR, methylene-THF reductase; CODH, carbon monoxide dehydrogenase/acetyl-CoA synthase; POR, pyruvate oxidoreductase; ICDHyr, isocitrate dehydrogenase; OORr, oxoglutarate oxidoreductase; FRD2, fumarate reductase; ME2, malic enzyme; NFN, electron-bifurcating transhydrogenase; HYDA_Ech/HYDFDN2/H2CYTBr/HYD-NADH, hydrogenases; FDXMQpp, complex I-like enzyme; RDase, reductive dehalogenase; ATPS, ATP synthase; WLP, Wood-Ljungdahl pathway; and TCA cycle, tricarboxylic acid cycle.

conserved domain patterns (Fig. 4). The Hyd-3 β-subunit corresponds with that of electron-bifurcating hydrogenases, though it possesses an extra NAD/FAD-binding domain similar to that of the *Sporomusa*-type NFN bifurcating oxidoreductase (25). The β-subunits of Hyd-4 and Hyd-6 align with the non-bifurcating hydrogenase of

*Syntrophomonas wolfei,* which are smaller and lack two FeS clusters compared to that of their electron-bifurcating counterparts (Fig. 4) (26, 27). These hydrogenases may also oxidize NADH to produce hydrogen—especially if NADH accumulates to high concentrations in the cell and the intracellular hydrogen concentration is low—and propose a mechanistic explanation for the consumption of excess electron equivalents in *Dehalobacter.*

Most Group A3 [FeFe] hydrogenases bifurcate electrons using equimolar amounts of NADH and reduced ferredoxin to produce hydrogen (29), which has been demonstrated in many organisms, including *Thermotoga maritima* (29), *Acetobacterium woodii* (30), *Moorella thermoacetica* (31), and *Ruminococcus albus* (32). Experimental evidence of non-bifurcating behaviour exists for [FeFe] hydrogenases in several species, such as *S. wolfei* (27), *Syntrophus aciditrophicus* (26), and *Solidesulfovibrio fructosivorans* (formerly *Desulfovibrio fructosovorans*) (33)*,* though the latter has only been characterized in the NADP-reducing direction. An [FeFe] hydrogenase from *Caldanaerobacter subterranus* (formerly *Thermoanaerobacter tengcongensis*) was also originally reported to produce hydrogen from NADH alone (34), but further investigation revealed its dependence on an additional electron acceptor (35).

When NADH-dependent hydrogen production (HYD-NADH) is added to the *Dehalobacter* models, both can produce biomass from DCM alone, without amino acid supplementation, indicating a successful NADH sink (Fig. S5; Table S11). Growth rates under this condition are the same for each strain (0.007 day$^{-1}$; Table S12).

The physiological feasibility of strategy B relies on a balance of environmental factors and redox cofactor availability. Under low partial pressures (<10 Pa [4 µM]), the redox potential of hydrogen is about −260 mV, rendering hydrogen production from NADH (E′ = −320 mV) thermodynamically feasible without confurcation (36). In SC05, hydrogen concentrations typically remain <6 µmol in 160 mL bottles of 100 mL culture (16). When considering the air-water partitioning coefficient of 51.7, this equates to an aqueous concentration of <2 µM. This maximum aqueous concentration remains sufficiently low to facilitate hydrogen production from NADH. In other biochemically characterized hydrogenases from *S. wolfei* and *S. aciditrophicus,* more hydrogen was produced when supplied a higher NADH/NAD$^+$ ratio, suggesting that NADH accumulation facilitates more efficient hydrogen evolution (26, 27); however, hydrogen production by the *S. aciditrophicus* hydrogenase has been reported at an NADH/NAD$^+$ ratio as low as 0.2 (26). The initially predicted redox imbalance in the *Dehalobacter* models stems from an accumulation of NADH, which strengthens the plausibility of strategy B *in vivo.*

## Strategy C: a combined approach most accurately describes *Dehalobacter* growth

Experimental yields from this study, as well as those interpreted from literature (15), were compared to the FBA-predicted yields for each model (Fig. 5). For strain SAD, during CF dechlorination (Mode 1) and DCM mineralization (Mode 2), the experimentally determined yields from this work align with the FBA-predicted yield in strategy B, while experimental yields from literature better align with strategy C (NADH-dependent hydrogenase ± amino acid supplementation). This difference may be due to the presence of different microbial community members at the time of each experiment. Amino acid supplementation alone does not increase the yields to match experimental values in Modes 1 and 2, even with increased rates of amino acids allowed. For strain DAD, however, the experimental yields align best with amino acid supplementation alone (Fig. 5).

Hydrogen export has been measured in SC05 (16), but is only predicted by the model when using an NADH-dependent hydrogenase (strategies B and C, Table S12), nominating this strategy as the most phenotypically accurate. Overall, a combined strategy using an NADH-dependent hydrogenase with additional carbon source supplementation (strategy C) may occur in SC05-UT to explain experimental hydrogen production by strain SAD and reflect its experimental growth yields. The strains' differential success

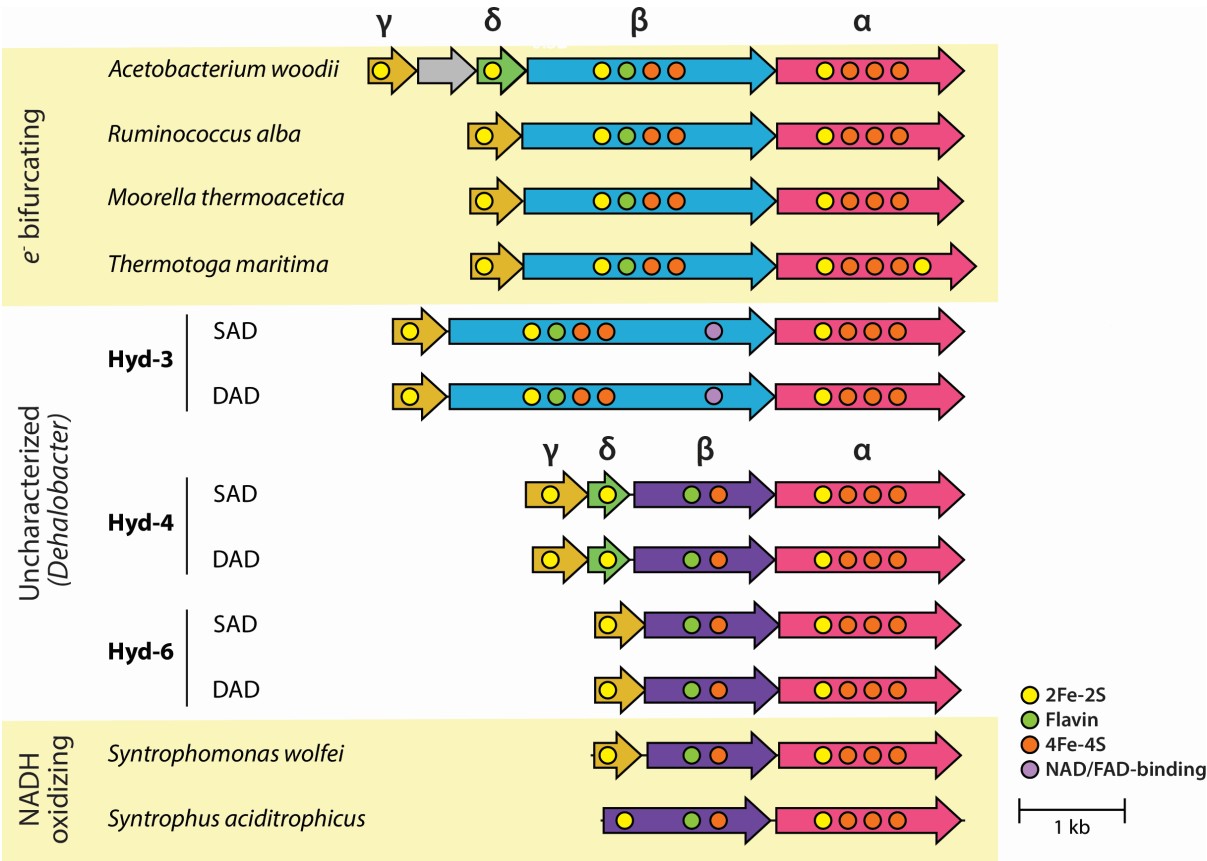

**FIG 4** Group A3 [Fe-Fe] hydrogenases in *Dehalobacter* strains SAD and DAD and their predicted conserved motifs compared to characterized $e^-$ bifurcating and NADH-oxidizing hydrogenases (27). Subunits are colored by homology, as clustered using clinker (28).

under DCM mineralization conditions in DCME may be influenced by increased access to amino acids and carbon sources, allowing strain DAD to grow faster than strain SAD. However, factors external to reaction stoichiometry, such as kinetic limitations, may also drive community composition in this ecosystem, as discussed below.

## Limitations of flux balance analysis and additional impacts on growth

In addition to reaction stoichiometry, several other factors may impact the growth of each strain, including resilience to toxicity (to CF especially) or membrane constraints (considering membrane-bound respiratory machinery), as well as enzyme kinetics and specificity. Though these considerations are incompatible with stoichiometric modeling, their potential to impact cell growth is briefly described below.

CF toxicity is known to impact many microbes, especially methanogens, acetogens, and others that require methyltransferases to catalyze key reactions—including other dechlorinating organisms (37–39). In *Dehalobacter*, CF dechlorination occurs in the periplasmic space, decreasing the impact of CF toxicity on its intracellular enzymes. Experimentally, we see dechlorination and growth in SC05-UT at aqueous CF concentrations reaching 1 mM (15, 16). Whether the two strains are differentially resilient to high concentrations of CF is not yet known, although we find no genomic or proteomic evidence for such a difference.

Cellular membrane constraints can limit cell growth, especially when membrane-bound enzymes control respiration (40–42). In *Dehalobacter*, its RDases and many of its hydrogenases are membrane-bound; thus, a larger surface area could facilitate the growth of one organism over another. *Dehalobacter* strains are very consistently sized

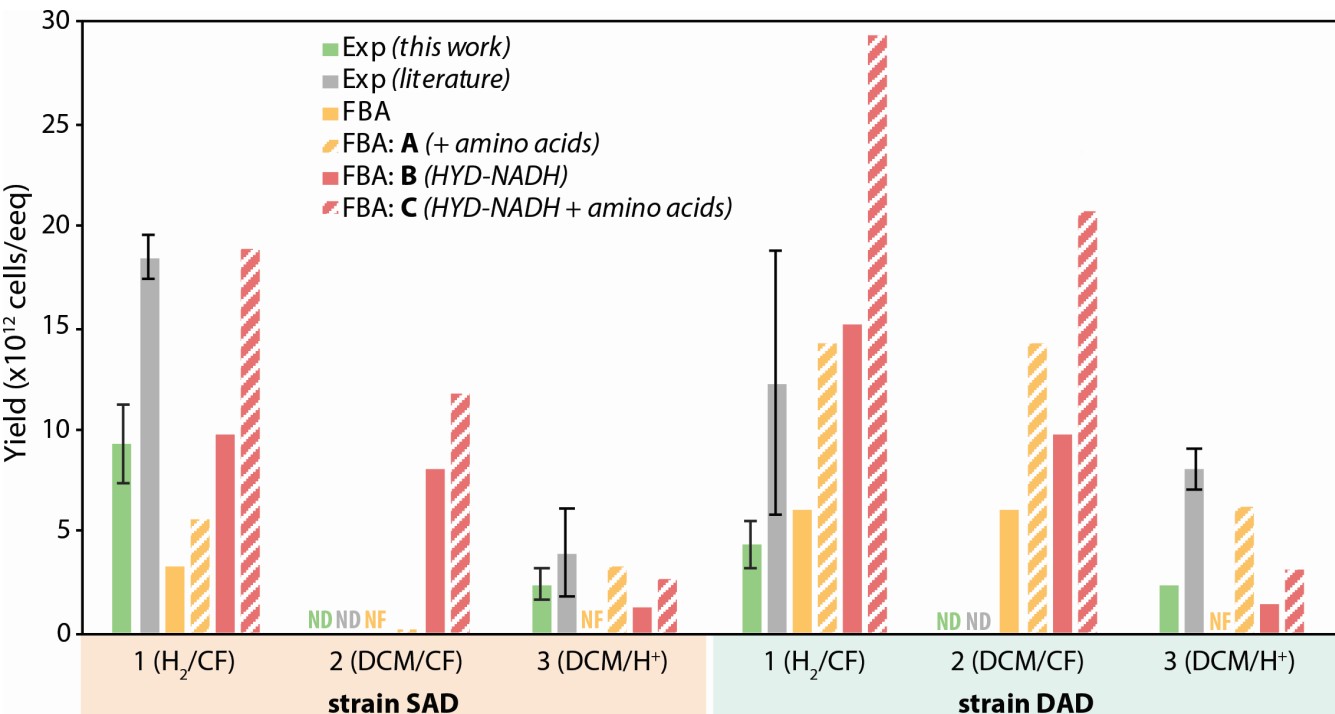

**FIG 5** Comparison of yields for each *Dehalobacter* strain during each growth mode, as determined experimentally and through flux balance optimization. ND, not determined; NF, not feasible. Error bars represent standard deviation for strain SAD and range for strain DAD.

(2–3 µm long, 0.3–0.5 µm in diameter [7, 43]), so membrane constraints are unlikely to be the main source of growth variations in these strains.

There are five single nucleotide variations (SNVs) between the homologous RDases in strains SAD and DAD (14), which may impact CF binding ($K_s$) or dechlorination rate ($\mu_{max}$). Heterologous expression of the expressed strain DAD RDase and subsequent enzyme kinetics could determine if these SNVs play a role in the *Dehalobacter* population dynamics in SC05. Similarly, there are sequence variations in several Mec cassette enzymes between strains SAD and DAD (14), and the development of enzyme assays to compare Mec cassette activity in each strain may shed more light on their DCM mineralization dynamics.

## Other possibilities for redox balancing

The architecture of the *Dehalobacter* electron transport chain is not fully elucidated. Though menaquinone has been crystallized between the RdhA and RdhB subunits of a *Dehalobacter* RDase (44), the role of the RdhC subunit is still unknown. Due to its structural similarity to subunits of the Nuo NADH:ubiquinone oxidoreductase (complex I), and its loose association with the other RDase subunits and ETC-related hydrogenases, it has been posited as an electron-transfer protein (4). This subunit may be able to accept electrons from intermediates like NAD(P)H or ferredoxins more directly, which would affect the redox balance of the cell, but further studies must be performed to further characterize this subunit's function.

## Intracellular electron transfer is predicted to mediate self-feeding during CF dechlorination

During regular culture maintenance, SC05-UT is fed CF alone without exogenous $H_2$ (Mode 2), during which electron cycling could occur two ways. The first is transmembrane hydrogen cycling, wherein a [NiFe] group 1a Hup-type periplasmic uptake hydrogenase (H2CYTBr) uses periplasmic $H_2$ to dechlorinate CF to DCM, which is then

mineralized to produce cytosolic $H_2$. After diffusion to the periplasm, this $H_2$ is re-oxidized by H2CYTBr for further dechlorination of CF (Fig. S5). FBA predicts transmembrane hydrogen cycling as the main electron transfer mechanism in strain SAD (Fig. 6A), and Hyd-2 (HYDA_Ech) and Hyd-8 (H2CYTBr) are the most consistently expressed by strain SAD under all conditions. Hyd-8 is also the most highly expressed hydrogenase by strain SAD.

In many simulations, HYDFDN is the main uptake hydrogenase. In this process, the DCM mineralization phase leads to the production of $H_2$ by HYDA_Ech or HYD-NADH, which accumulates until the second phase, wherein the hydrogen pool is recycled by HYDFDN to donate $e^-$ back to CF (Fig. 6B). In these instances, $H_2$ can be shuttled through a recycling loop between HYDA_Ech ($H_2$ producing), HYD-NADH ($H_2$ producing), and HYDFDN ($H_2$ consuming) (Fig. 6B). Notably, FBA models do not inherently account for temporal concentration dynamics (i.e., the consumption of hydrogen by HYDFDN may first require the accumulation of hydrogen in the system), so simulations were also performed with $H_2$ consumption by HYDFDN inhibited (Table S11, solutions 17–22). In these simulations, transmembrane hydrogen cycling only occurs in strain SAD (strategy A: with amino acids and no NADH-dependent hydrogenase); all other simulations report intracellular electron transfer using reduced ferredoxins that were produced from DCM assimilation to reduce menaquinone (FDXMQpp) and subsequently dechlorinate CF. Future work could perform flux coupling analysis to highlight reaction interdependencies and further cement the role of hydrogenases in electron cycling (45, 46).

## Model-predicted hydrogen cycling mirrors other domains of life

During CF dechlorination and DCM mineralization, both *Dehalobacter* models predict proton translocation through its Ech-type hydrogenase (HYDA_Ech) or the complex I-like oxidoreductase (FDXMQpp) (Fig. 6). Both necessitate a source of reduced ferredoxins, whose production is predicted via a hydrogenase cascade (Fig. 7). The model-predicted transmembrane hydrogen cycling (Fig. 6A) mirrors the hydrogen cycling model that was first proposed to explain transient $H_2$ observed in batch culture of *Desulfovibrio* spp. (Fig. 7B) (47). In this model, a cytoplasmic hydrogenase converts electrons and protons generated during lactate fermentation to $H_2$, which subsequently diffuses to the periplasm. A periplasmic hydrogenase re-oxidizes $H_2$, establishing a proton gradient that drives ATP production. Simultaneously, electrons flow through several membrane-bound electron transfer complexes to ultimately reduce sulfate. This model has since expanded to include additional hydrogenases and $H_2$-independent electron transfer to the electron transport chain via the complex I-like oxidoreductase or unknown electron transfer complexes (like Fig. 7B) (33, 48–51).

Similarly, in *Methanosarcina barkeri*, the role of $H_2$ cycling in energy conservation has been determined experimentally (Fig. 7C) (55, 56). Here, $H_2$ is produced by Ech and Frh (a F420-oxidizing cytoplasmic hydrogenase), which diffuses to the periplasm and is re-oxidized by Vht (a quinone-reducing periplasmic hydrogenase) to form a proton gradient harnessed by ATP synthase. An analogous system also exists in *Methanosarcina mazei* (57) and resembles *Dehalobacter* transmembrane $H_2$ cycling. The common occurrence of both cytoplasmic and periplasmic hydrogenases among diverse microorganisms suggests that this mechanism is widespread in nature, especially among anaerobes (55).

Without a periplasmic hydrogenase, like in fermenting *Pyrococcus* and *Thermococcus* strains, cytosolic $H_2$ is produced by Ech and later re-oxidized by soluble [NiFe] hydrogenases to regenerate redox cofactors from what is otherwise a waste product (58, 59). This temporal intracellular hydrogen recycling also occurs in *Solidisulfovibrio fructosivorans* during fermentation, where a bidirectional [FeFe] electron-bifurcating hydrogenase (Hnd) reversibly produces hydrogen as an electron sink when grown under fermentative conditions and re-consumes it under respiratory conditions when an alternative electron acceptor is available (60). This mirrors simulations of sequential DCM mineralization (Mode 3) and subsequent CF reduction upon refeeding (Mode 1) by *Dehalobacter*.

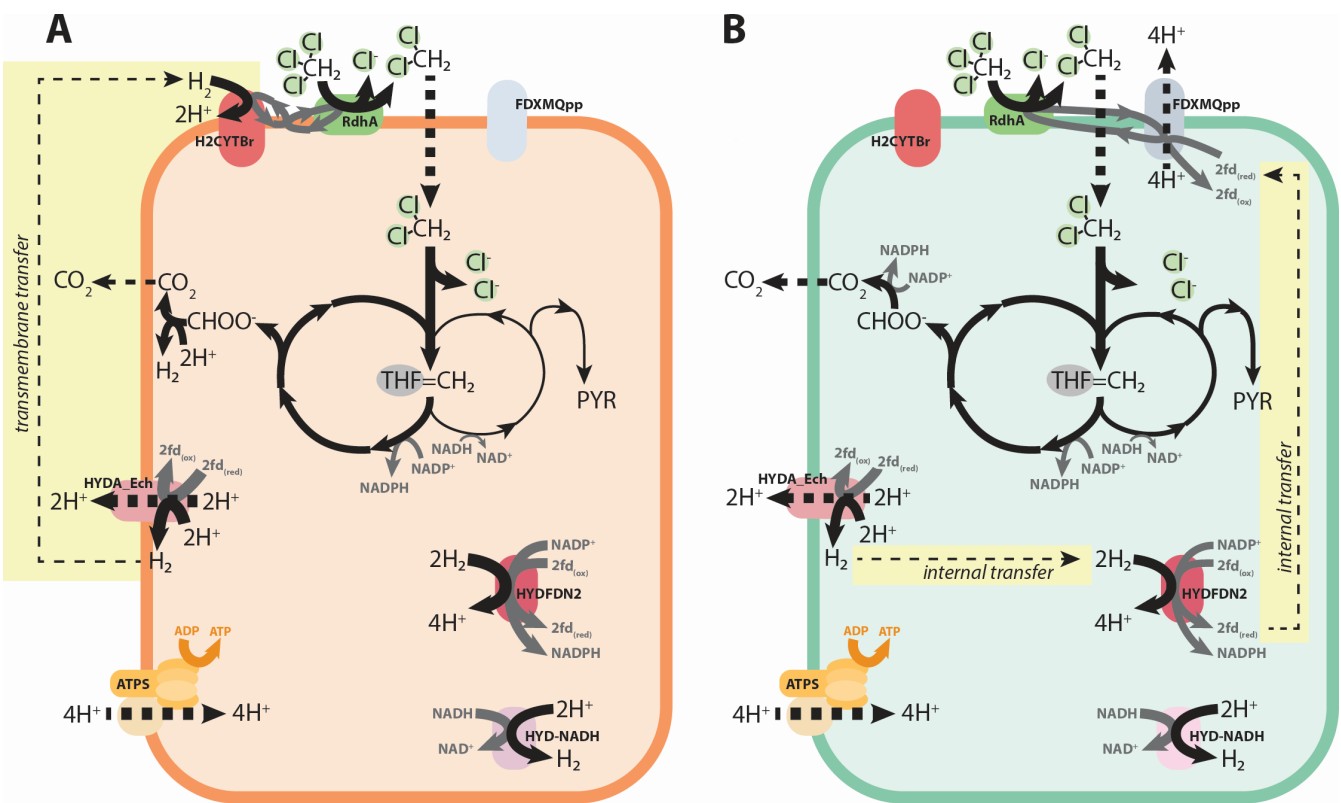

**FIG 6** FBA-predicted electron transfer in *Dehalobacter* strains, including (A) transmembrane transfer and (B) internal transfer. Arrows are weighted by relative flux. Metabolites and enzymes are abbreviated as follows: THF, tetrahydrofolate; fdx, ferredoxin; HYDA_Ech, Ech-type hydrogenase; HYDFDN, electron-bifurcating hydrogenase; H2CYTBr, cytochrome B-reducing hydrogenase; HYD-NADH, NADH-dependent hydrogenase; FDXMQpp, complex I-like enzyme; RDase, reductive dehalogenase; and ATPS, ATP synthase.

## Organohalide syntrophy vs. competition

Though FBA simulations are helpful for determining the metabolic capability of each strain alone, in the SC05-UT and DCME subcultures, both strains have co-existed to some extent for years. These strains were initially expected to interact symbiotically through intercellular compartmentalization, with labor divided between strains such that one specializes in CF dechlorination, producing DCM for the other strain, which performs DCM mineralization and produces $H_2$ for the CF-dechlorinator. Sequential steps of other pollutant biotransformation pathways that produce inhibitory intermediates are often compartmentalized within different cells across the population (61–63). This scheme reduces metabolic burden, maximizes the community growth rate, and may theoretically circumvent the buildup of toxic intermediates (64). Experimentally, however, some DCM accumulates during CF dechlorination in SC05, and enzymes specific to both steps in the pathway are expressed by both *Dehalobacter* strains, regardless of whether steps are occurring (i.e., the CF-specific RDase is still expressed by strain DAD in DCME after extended enrichment on DCM [Table S9]). Therefore, this scheme does not accurately describe the population dynamics of SC05.

An alternate scheme poses these strains as competitors for CF and DCM, with internal cycling of electrons in each strain or a shared pool of $H_2$ as an electron donor to continually reduce CF completely to carbon dioxide. Complete use of one substrate by a single organism theoretically maximizes the yield of the community, reducing the need for redundant cell maintenance expenditures (64). The need to avoid the buildup of toxic intermediates may not play a large role in these organisms, since CF dechlorination occurs extracellularly in the periplasm, which allows the cell to control import of DCM for intracellular mineralization. Experimentally, each strain is capable of both CF and

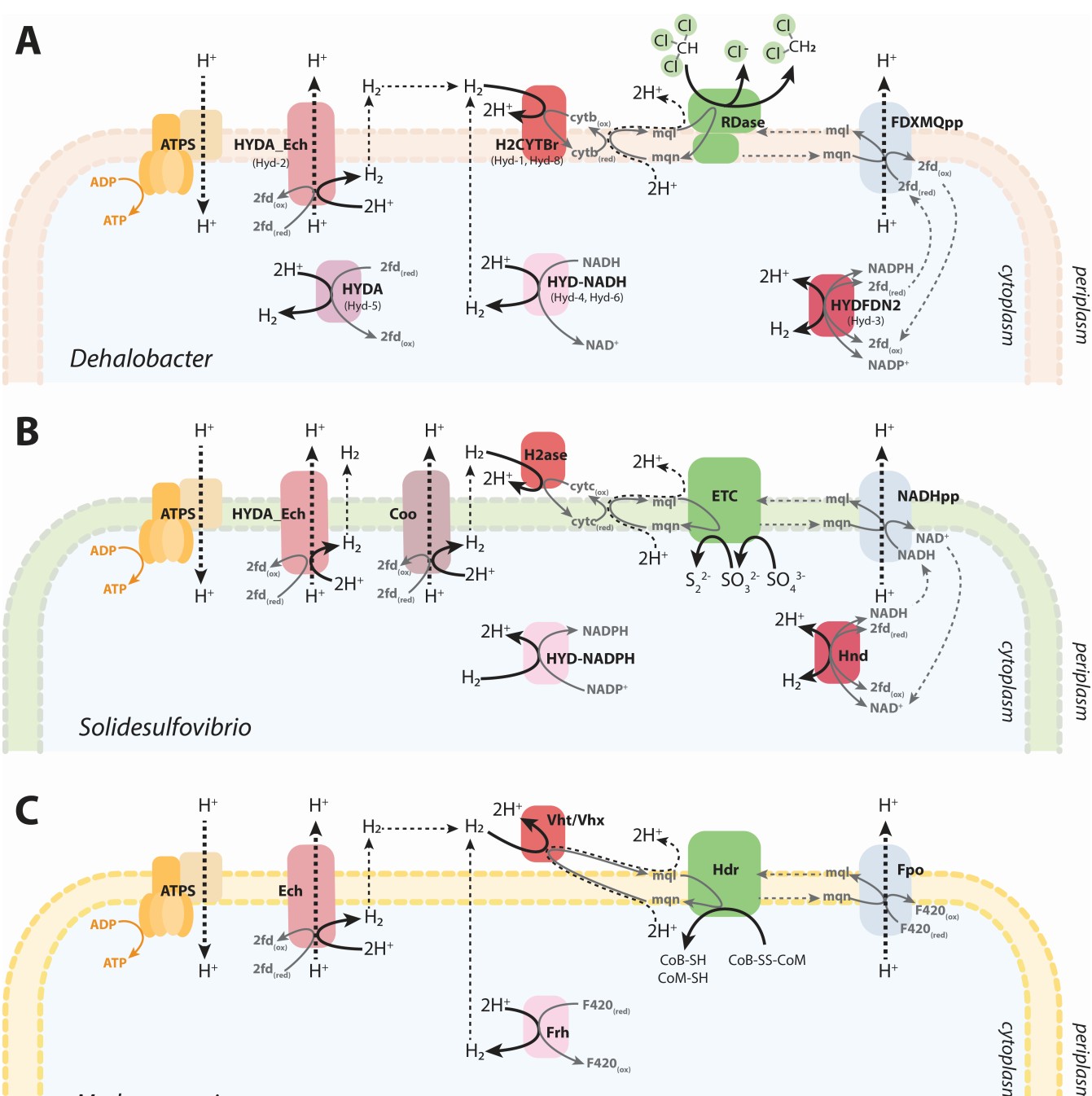

**FIG 7** Proposed schematic for intracellular hydrogen cycling (without fluxes or stoichiometry) in (A) *Dehalobacter* compared to characterized strains: (B) *Solidesulfovibrio* (52) and (C) *Methanosarcina* (53, 54). Metabolites are abbreviated as follows: fdx, ferredoxin; mql, menaquinol; mqn, menaquinone; cytb, cytochrome b; CoB, coenzyme B, and CoM, coenzyme M. Enzymes are abbreviated as follows: H2ase, hydrogenase; HYDA_Ech, Ech-type hydrogenase; HYDFDN, electron-bifurcating hydrogenase; H2CYTBr, cyt B-reducing hydrogenase; HYD-NADH, NADH-dependent hydrogenase; NADHpp/FDXMQpp, complex I-like enzyme; RDase, reductive dehalogenase; ATPS, ATP synthase; and ETC, electron transfer proteins.

DCM utilization, and we typically see only one strain thriving in each culture. Furthermore, metabolic modeling suggests that higher yields are feasible where each strain is consuming both CF and DCM (Mode 2), as predicted yields for DCM mineralization alone (Mode 3) are low (Fig. 5). Our data suggest that in the SC05-UT and DCME enrichment cultures, these strains are more competitive than cooperative (Fig. S8).

Despite typically finding only one dominant strain in each culture, there is some evidence for tandem degradation under some circumstances in SC05-UT replicate 2 (Fig. 1). In this replicate, strain DAD did not grow, while CF was converted to DCM, but suddenly flourished (100-fold increase in abundance) as DCM was degraded. In this instance, it seems that strain SAD dechlorinated CF to DCM, which strain DAD then mineralized, perhaps while exchanging amino acids (e.g., ASPT in strain DAD supplying nitrogen intermediates to strain SAD) or cycling $H_2$ (i.e., $H_2$ production by strain DAD fueling the CF reduction by strain SAD). This suggests that organohalide syntrophy between the two strains is also feasible, likely dependent on the environmental conditions in the culture, such as solvent concentration, hydrogen partial pressures, and amino acid availability.

Notably, the *Dehalobacter* population in the parent culture used for bioremediation (fed a more complex electron donor mixture of methanol and ethanol, and a lower concentration of CF) consists of a lower SAD:DAD ratio of ~80:20. This environment may be more conducive to cooperation than the highly enriched conditions of SC05-UT and DCME. Lower electron acceptor concentrations and more diverse substrates are also a better representation of contaminated groundwater, which is important for bioremediation, where mixed consortia drive pollutant degradation. Future work should test co-culture dynamics, model metabolite exchange, or use proteomics to identify shared pathways in the original parent culture, which could reveal cooperative strategies for optimizing remediation.

## Role of methanogens in SC05

The feasibility of DCM mineralization depends on environmental hydrogen concentrations, likely due to enzyme inhibition of hydrogen-evolving hydrogenases (65–67). DCM mineralizers require a syntrophic partnership with hydrogenotrophs like *Solidesulfovibrio* spp. (68) or methanogens (65) to keep the hydrogen concentration low enough for DCM degradation to proceed.

Hydrogenotrophic methanogens are prominent members of SC05-UT and DCME, producing methane from hydrogen and carbon dioxide. These methanogens are an interesting partner for *Dehalobacter*, as they both compete and cooperate with *Dehalobacter* depending on its mode of metabolism. DCM mineralization consistently coincides with methane production, suggesting cooperation between methanogens and *Dehalobacter*, where hydrogen consumption by methanogens reverses inhibition of the hydrogenases that are required for DCM mineralization (Fig. 8B).

During CF dechlorination, however, *Dehalobacter* also uses hydrogen as an electron donor. Dechlorination alone can maintain a sufficiently low hydrogen concentration for DCM mineralization, reducing the need for active methanogens. Several ecological advantages allow *Dehalobacter* to successfully compete with methanogens for hydrogen under these conditions. CF dechlorination is thermodynamically feasible at lower hydrogen concentrations than methanogenesis (Fig. S8). Additionally, CF itself inhibits methanogenesis at relatively low concentrations, which further allows *Dehalobacter* to outcompete methanogens for hydrogen.

Despite competition for hydrogen as an electron donor, the presence of methanogens is still critical to ecosystem health in CF-fed SC05-UT. Twice as much hydrogen is produced from DCM mineralization as is consumed by dechlorination of an equimolar concentration of CF to DCM (15). This excess of hydrogen necessitates additional consumption by methanogens after CF has been completely dechlorinated to DCM and *Dehalobacter* shifts to DCM mineralization alone. These methanogens provide a critical hydrogen sink to remove excess electrons from the *Dehalobacter* cycle as methane. Future work could simulate this series of inter- and intra-species interactions using dynamic modeling approaches like dFBA (69), or co-simulations with methanogen models using community gap-filling (13).

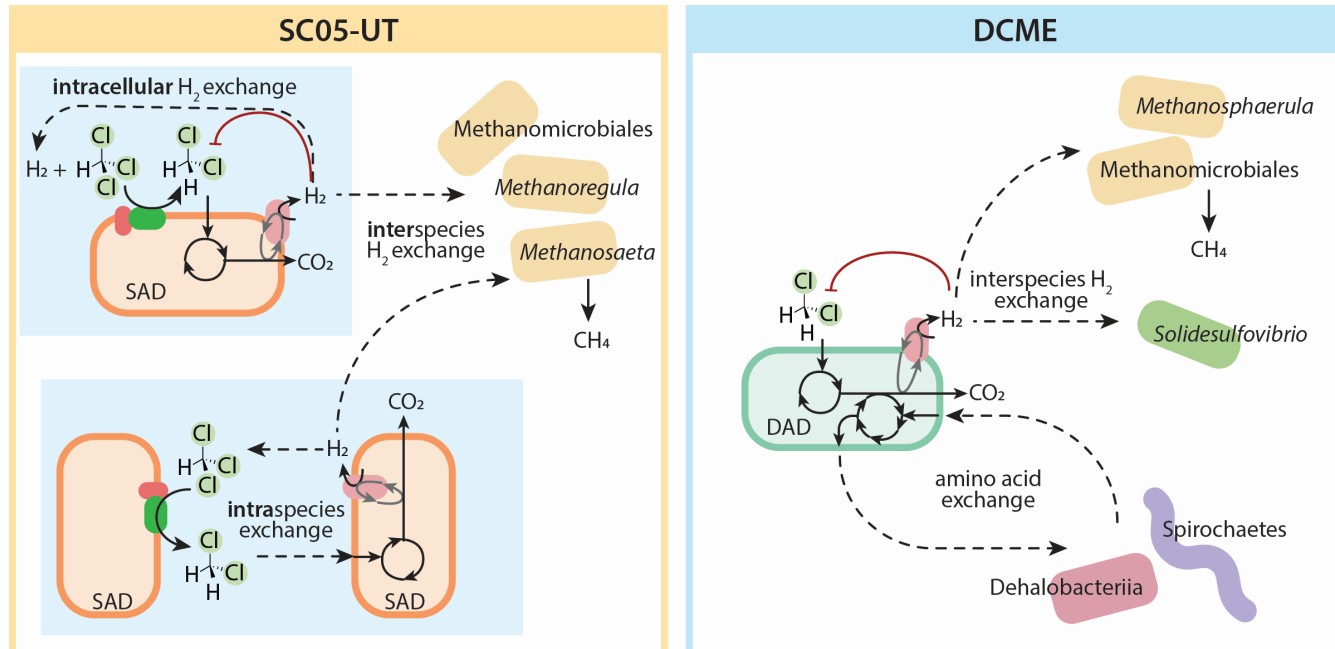

**FIG 8** Proposed model of redox imbalance resolution of (A) *Dehalobacter restrictus* SAD in SC05-UT and (B) *Ca*. Dehalobacter alkaniphilus DAD in DCME, considering all data. Metabolite exchanges are depicted with dashed arrows; inhibition is depicted in red.

## Significance for bioremediation

Due to lack of microbial isolates, the metabolic mechanisms of key microbes in the CF and DCM–degrading SC05 culture have remained elusive despite its wide use in bioaugmentation. By closing two distinct *Dehalobacter* genomes and reconstructing metabolic models of each, we can better propose possible mechanisms of energy and biomass production during the multiple growth cases we observe experimentally. Moreover, studying the redox limitations of the pathways in each strain provides insight into caring for the cultures and an explanation of its many possible growth phenotypes.

More broadly, the differential ability of *Dehalobacter* strains SAD and DAD to produce biomass from DCM without CF dechlorination is a case study for the impact of seemingly unrelated metabolic genes on a pathway. Despite having similar genomes and sharing genes for key enzymes, differential use of several genes allows more efficient redox balancing and more growth in strain DAD. These results have implications for *in situ* biomarker tracking, as detection of the *mec* cassette from strain SAD *in situ* would not necessarily portend efficient DCM remediation.

## MATERIALS AND METHODS

### Model reconstruction and curation

Model construction is expanded upon in Text S1. Genome-scale metabolic models representing *Dehalobacter* strains SAD (*i*OB638) and DAD (*i*OB649) were drafted from model *i*HH623, a previously curated *Dehalobacter* model (12, 13). The reactions and compounds in the template model were converted to be in accordance with the BiGG database. The biomass equation was not changed from previous model construction and includes 50% protein (using an amino acid composition similar to *Bacillus subtilis*, a Gram-positive microbe with definitive composition tests (70)), 10% RNA, 5% DNA, 5% phospholipids, 25% peptidoglycan, and 5% ash (12). A sensitivity analysis was performed to determine the influence of biomass composition on growth yields (Text S1.2; Fig. S1).

A comparison of metabolic genes in the genomes of *Dehalobacter* strains SAD and DAD is expanded upon in Text S2. These genomes were assembled as previously

described (18) and annotated with MetaErg (71). Metabolic pathways were reconstructed with KEGG Mapper (72) and validated against the genome annotations to curate the models. Genes with metabolic annotations not included in the initial draft reconstruction were reviewed for inclusion using PaperBLAST (73). RDase and transport reactions were updated to reflect experimentally determined electron donors and acceptors. All putative hydrogenases were classified with HYDdb (24), and reactions were adjusted accordingly. Added or changed reactions are summarized in Table S4. Models are summarized in Table 2. Available enzyme cofactors are listed in Text S2.4 (Table S8).

## Thermodynamic constraints

Additional curation was performed to account for thermodynamic constraints for the three electron donor/acceptor pairs used by SC05 (Text S1). Briefly, the $\Delta G^{\circ\prime}$ of each reaction is calculated using equation 1, where $\Delta G^{\circ\prime}_a$ and $\Delta G^{\circ\prime}_d$ are determined from the energies of formation of each compound in each half reaction assuming standard conditions (74). The maximum ATP/$e^-$ ratio is determined with equation 2 assuming $\Delta G'_p = 50$ kJ/ mol ATP, where $n$ is the number of electrons transferred in the reaction. The maximum H$^+$/$e^-$ ratio—the number of protons translocated across the membrane per mol $e^-$ transferred from donor to acceptor—is calculated by equation 3, assuming four protons are required per ATP formed by ATP synthase (75). The proton translocation stoichiometry in the model was adjusted such that when non-growth associated maintenance (NGAM) is maximized during flux balance analysis (FBA), the ATP/$e^-$ and H$^+$/$e^-$ ratios fall within the thermodynamic constraints for each mode of metabolism (set at 0.5 and 2, respectively; Text S1.3). Energy transfer efficiencies were estimated by comparing theoretical and experimental values (Table S3).

$$\Delta G^{\circ\prime}_{rxn} = \Delta G^{\circ\prime}_a - \Delta G^{\circ\prime}_d \quad (1)$$

$$\left(\frac{n_{ATP}}{n_{e-}}\right)_{max} = \frac{\Delta G^{\circ\prime}_{rxn}}{n\Delta G'_P} \quad (2)$$

$$\left(\frac{n_{H^+}}{n_{e-}}\right)_{max} = 4\left(\frac{n_{ATP}}{n_{e-}}\right)_{max} \quad (3)$$

These parameters and experimental decay rates were used to calculate growth associated maintenance (GAM) and NGAM, which were ultimately set to 60 mmol ATP gdw$^{-1}$ and 3.6 mmol ATP gdw$^{-1}$ day$^{-1}$, respectively (Text S1.4). Substrate uptake rates were constrained to experimentally relevant values (10 mmol gdw$^{-1}$ day$^{-1}$) (74).

## Proteomic searching of *Dehalobacter* strains SAD and DAD

Reanalysis of pre-collected proteomic data (14, 76) was performed using the closed *Dehalobacter* strains SAD and DAD genomes (NCBI accessions: CP148031, CP148032 [18]). A description of protein extraction and mass spectrometry is described in our related work (14). All MS/MS samples were processed through X! Tandem (the GPM, thegpm.org; version X! Tandem Aspartate [2020.11.12.1]) for initial identification of peptides, and further analyzed through Scaffold (version Scaffold_5.3.0, Proteome Software Inc., Portland, OR), as described below.

X! Tandem was set up to search the custom database (6,683 entries), assuming the digestion enzyme trypsin. X! Tandem was searched with a fragment ion mass tolerance of 0.40 Da and a parent ion tolerance of 20 PPM. Carbamidomethyl of cysteine and selenocysteine was specified in X! Tandem as a fixed modification. Cyclization of N-terminal glutamate and glutamine, ammonia loss of the N-terminus, deamidation of asparagine and glutamine, and oxidation or dioxidation of methionine and tryptophan were specified in X! Tandem as variable modifications.

Scaffold was used to validate MS/MS-based peptide and protein identifications. Peptide identifications were accepted if they could be established at greater than 95.0% probability by the Peptide Prophet algorithm (77) with Scaffold delta-mass correction (peptide FDR = 1.1%). Protein identifications were accepted if they could

mSystems

be established with greater than 90.0% probability and contained at least one identified peptide (protein FDR = 0.6%). Protein probabilities were assigned by the ProteinProphet algorithm (78). Proteins that contained similar peptides and could not be differentiated based on MS/MS analysis alone were grouped to satisfy the principles of parsimony.

## Flux balance analysis

Flux balance analysis (FBA) was performed using COBRApy (79), with the biomass reaction as the objective function. FBA was conducted for each model individually. Constraints were applied to each model to render the simulations more physiologically relevant, considering several experimentally derived data sets: proteomic expression data (Data set S2 at https://doi.org/10.6084/m9.figshare.26972656), experimental growth yields, and dechlorination rates (15). The FBA code and models can be found in the following GitHub repository: https://github.com/LMSE/Dehalobacter_modelling.

## Strain detection in each culture

Sub-transfers of SC05-UT and DCME were weekly set up as previously described (15). Briefly, three sub-transfers of SC05-UT and two sub-transfers of DCME were amended with 25 µmol CF (0.25 mM CF aqueous) and 10 µmol hydrogen (1 mL $H_2/CO_2$, 20:80, vol/vol) to jump-start degradation. The SC05-UT replicates were amended with another 50 µmol CF on day 4. Each bottle's headspace was analyzed by gas chromatography every three to five days, as previously described (15), and 1 mL of culture was sampled for DNA extraction and qPCR.

DNA was extracted from 1 mL samples by pelleting the culture via centrifugation at 13000 × $g$ for 15 minutes and then using the KingFisher Duo Prime (Thermo Scientific, Waltham, MA) and MagMAX Microbiome Ultra Nucleic Acid Isolation Kit (Applied Biosystems, Waltham, MA) to extract from the pellets, as specified in the manual but modified to include the entire volume of sampled culture; therefore, a 100% DNA extraction efficiency is assumed for culture samples.

Each strain of *Dehalobacter* was quantified using primers previously designed for a core gene in *Dehalobacter* with high sequence variability: flagellar basal-body rod protein, *flgC* (5). Using these primers, quantitative PCR (qPCR) was performed on DNA samples taken during previous time course experiments with the SC05-UT and DCME subcultures (15). The qPCR reaction mixtures were prepared in a UV-treated PCR cabinet (ESCO Technologies, Hatboro, PA) and contained 10 µL of 2× SsoFast SYBRGreen (Bio-Rad, Hercules, CA), 0.25 µM of each *flgC* primer, and 2 µL of template DNA. The amplification program and analyses were conducted using a CFX96 Touch Real-Time PCR Detection System and the CFX Manager software (Bio-Rad). The qPCR method included an initial denaturation step at 98°C for 2 minutes, followed by 40 cycles of 5 se at 98°C and 10 s at 55°C, with a 2°C $s^{-1}$ ramp between temperatures. Quantification was performed using 10-fold serial dilutions of PCR-produced standard DNA, amplified from a larger region of the *flgC* gene, which was quantified using the Qubit dsDNA High Sensitivity Assay Kit and Qubit Fluorometer (Thermo Scientific).

## ACKNOWLEDGMENTS

The authors would like to thank Robert Flick for his aid in data acquisition via mass spectrometry.

Funding for this work was provided by the Natural Science and Engineering Research Council (NSERC) though a Discovery Grant to E.A.E. and Doctoral Scholarships to O.B., as well as a Genome Canada Bioinformatics and Computational Biology (BCB project 285MPR) subgrant to E.A.E. and R.M. We also acknowledge funding from the Canada Research Chairs Program to R.M. and E.A.E. The funders had no role in study design, data collection, or interpretation.

## AUTHOR AFFILIATION

[1]Department of Chemical Engineering and Applied Chemistry, University of Toronto, Toronto, Ontario, Canada

## AUTHOR ORCIDs

Olivia Bulka ⓘ http://orcid.org/0000-0003-1691-6892
Elizabeth A. Edwards ⓘ http://orcid.org/0000-0002-8071-338X
Radhakrishnan Mahadevan ⓘ http://orcid.org/0000-0002-1270-9063

## FUNDING

| Funder | Grant(s) | Author(s) |
| --- | --- | --- |
| Natural Sciences and Engineering Research Council of Canada | PGS-D | Olivia Bulka |
| Genome Canada | BCB project 285MPR | Elizabeth A. Edwards |
| | | Radhakrishnan Mahadevan |
| Natural Sciences and Engineering Research Council of Canada | | Elizabeth A. Edwards |
| Canada Research Chairs | | Elizabeth A. Edwards |
| | | Radhakrishnan Mahadevan |

## AUTHOR CONTRIBUTIONS

Olivia Bulka, Conceptualization, Data curation, Formal analysis, Investigation, Methodology, Visualization, Writing – original draft, Writing – review and editing | Elizabeth A. Edwards, Conceptualization, Supervision, Writing – review and editing | Radhakrishnan Mahadevan, Conceptualization, Methodology, Supervision, Writing – review and editing

## ADDITIONAL FILES

The following material is available online.

### Supplemental Material

**Supplemental material (mSystems00847-25-s0001.pdf).** Supplemental texts and figures, links to external data sets, and Tables S1-S7.
**Supplemental Tables (mSystems00847-25-s0002.xlsx).** Tables S9-S12.

### Open Peer Review

**PEER REVIEW HISTORY (review-history.pdf).** An accounting of the reviewer comments and feedback.

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
