## [Reviewer comments · mSystems]

Modeling reveals metabolic basis of competition among *Dehalobacter* strains during tandem chloroform and dichloromethane metabolism

Olivia Bulka, Elizabeth Edwards, and Radhakrishnan Mahadevan

Corresponding Author(s): Radhakrishnan Mahadevan, University of Toronto

Review Timeline:

Submission Date:	June 10, 2025
Editorial Decision:	June 21, 2025
Revision Received:	June 24, 2025
Accepted:	June 25, 2025

Editor: Saheed Imam

Reviewer(s): The reviewers have opted to remain anonymous.

Transaction Report:

DOI: <https://doi.org/10.1128/msystems.00847-25>

Re: mSystems00847-25 (Modeling reveals metabolic basis of competition among *Dehalobacter* strains during tandem chloroform and dichloromethane metabolism)

Dear Dr. Radhakrishnan Mahadevan:

While the authors have adequately addressed most of the reviewer concerns, I still feel there is a lack of clarity around the data provided in the original Table 4 (now Figure 5). In your response to Reviewer #2, you state that this error was due to incorrect units of 16S copies/eq donor, instead of cells/eq donor and that there are three 16S copies in strain DAD and four copies in strain SAD, resulting in the numbers being 3 to 4-fold high. However, the numbers in the updated table do not reflect these values. For instance, in the original Table 4 for the DAD mode 1, a value of 37.1 {plus minus} 7.8 was given in the original table as data generated in this study, while in the new table this is now 4.4 {plus minus} 1.2. That's an 8.4-fold reduction that doesn't correspond to 3 copies of 16S copies in strain DAD. There's a similar trend for other numbers in the table, where the new numbers are far lower (see SAD mode 1) or higher than I would have expected if this was just based on the 16S copies. I'm I missing something here? Please clarify this and/or provide updated numbers that reflect this transformation.

Revision Guidelines

Sincerely,
Saheed Imam
Editor
mSystems

Re: mSystems00847-25R1 (Modeling reveals metabolic basis of competition among *Dehalobacter* strains during tandem chloroform and dichloromethane metabolism)

Dear Dr. Radhakrishnan Mahadevan:

Your manuscript has been accepted, and I am forwarding it to the ASM production staff for publication. Your paper will first be checked to make sure all elements meet the technical requirements. ASM staff will contact you if anything needs to be revised before copyediting and production can begin. Otherwise, you will be notified when your proofs are ready to be viewed.

Sincerely,
Saheed Imam
Editor
mSystems